# Process Independence Testing in Proximal Graphical Event Models

**Debarun Bhattacharjya**              DEBARUNB@US.IBM.COM

**Karthikeyan Shanmugam**       KARTHIKEYAN.SHANMUGAM2@IBM.COM

**Tian Gao**                           TGAO@US.IBM.COM

**Dharmashankar Subramanian**       DHARMASH@US.IBM.COM

*IBM Research, Thomas J. Watson Research Center, Yorktown Heights, NY, USA*

**Editors:** Bernhard Schlkopf, Caroline Uhler and Kun Zhang

## Abstract

Datasets involving irregular occurrences of different types of events over the timeline are increasingly commonly available. Proximal graphical event models (PGEMs) are a recent graphical representation for modeling relationships between different event types in such datasets. Existing algorithms for learning PGEMs from event datasets perform poorly on the task of structure discovery, which is particularly important for causal inference since the underlying graph determines the effect of interventions. In this paper, we explore causal semantics in PGEMs and study process independencies implied by the graphical structure of the model. We introduce (conditional) process independence tests for causal PGEMs, deploying them using variations of constraint-based structure discovery algorithms for Bayesian networks. Through experiments with synthetic and real datasets, we show that the proposed approaches are better at balancing precision and recall, demonstrating improved F1 scores over state-of-the-art baselines.

**Keywords:** Event datasets, Structure discovery, Event models, Graphical event models, Process independence, Local independence, Constraint-based methods

## 1. Introduction and Related Work

Causal discovery is of great interest in artificial intelligence, machine learning and statistics, and indeed in the broader realm of scientific discovery. While data obtained from intervening in a system and then measuring the ramifications is typically the gold standard for causal inference, it can be impractical to design and/or expect such data in many practical situations. As a result, causal discovery from purely observational data receives widespread attention in the literature and in practice across various domains.

Pearl (2009) proposed the framework of graphical models, particularly Bayesian networks, as a representation for causal analysis. By enforcing causal semantics on directed acyclic graphs describing conditional independence relationships between variables, Pearl developed an elegant mathematical theory that could analyze complex causal situations from simple probabilistic rules.

While causal Bayesian networks are powerful representations, they do not adequately capture temporal aspects, which can be key to effective causal modeling in practice. Early work on explicitly including time includes counterfactual/potential outcome approaches, such as by Robins (1994) on the structural nested mean model, along with approaches more aligned with classical statistics that are formulated as models on actual observations, such as Granger causality for time series (Granger, 1969). Graphical representations were later developed for representing discrete-time temporal processes, including dynamic Bayesian networks (Dean and Kanazawa, 1989; Murphy, 2002) and graphs for time series (Eichler, 1999).

In many domains, it is more common to observe irregular occurrences of 'events' rather than regular measurements that are typical in time series data. This is the case in clinical medicine and epidemiology, system maintenance, retail, politics, and numerous other applications. Such data is better represented by continuous time models. *Graphical event models* (GEMs) (Didelez, 2008; Gunawardana and Meek, 2016) are representations for marked (or multivariate) point processes for continuous-time event occurrences. They capture dependencies between various types of events over time, providing a framework that generalizes many parametric temporal models, including continuous time Bayesian networks (Nodelman et al., 2002), Poisson networks (Rajaram et al., 2005), Poisson cascades (Simma and Jordan, 2010), piecewise-constant conditional intensity models (Gunawardana et al., 2011), and forest-based point processes (Weiss and Page, 2013), among many others.

While GEMs are a useful high-level framework, it is necessary in practice to make specific assumptions about historical dependencies to actually learn a model from a real-world dataset. Any GEM needs to be specified in terms of the exact manner in which historical occurrences affect the rate at which an event occurs. Proximal graphical event models (PGEMs) have been proposed recently as a GEM where only the recent history determines the rate at which an event occurs (Bhattacharjya et al., 2018). As opposed to the more general PCIM (Gunawardana et al., 2011), they do not require domain knowledge if it is unavailable and avoid over-fitting during learning.

However, a major disadvantage of the state-of-the-art learner for PGEMs – a score-based method based on the Bayesian information criterion (BIC) – is its relatively poor performance on structure discovery when data is limited. As we will show through experiments, the learned graph from this prior work is often extremely sparse and fails to discover many true parents in the case of limited data. Most prior work on learning specific parametric graphical event models as well as closely related graphical representations use score-based approaches (Nodelman et al., 2003; Gunawardana et al., 2011; Bhattacharjya et al., 2020a,b,c). These approaches are generally inspired by score-based methods for Bayesian networks (Chickering, 2002).

Here we consider PGEMs through a causal lens and follow Didelez (2008) in exploring *process independence* in marked point processes associated with PGEMs. Process independence is a notion of independence pertaining to systems exhibiting temporal dynamics; variations of this idea have been studied previously and subsequently (Schweder, 1970; Meek, 2014; Mogensen et al., 2018). PGEMs are graphical representations that are particularly suitable for causal modeling of multivariate event streams, as the underlying assumption around the pertinence of recent occurrences often approximates the nature of causal influences in the real world. Process independence in causal PGEMs provides a useful avenue for learning such models from data, analogous to *constraint-based methods* for causal Bayesian networks (Spirtes et al., 2001). Constraint-based methods have been shown to have several advantages that are well known in the Bayesian network literature, including around handling confounders. Our intuition is that conducting statistical tests for process independence could provide similar benefits while doing multivariate conditioning in event streams.

While there is plenty of literature in statistics on determining dependence between two point processes (Perkel et al., 1967; Brillinger et al., 1976; Doss, 1989), there is hardly any literature on the sort of multivariate conditioning that is required for GEMs. Meek (2014) discussed the promise of testing for process independence in GEMs but assumed the availability of an oracle tester that verified process independence statements. We are unaware of prior work on relevant testers in this space, and therefore any prior empirical demonstrations in this regard. Furthermore, we note that while it may be possible to adapt testers for atemporal (i.i.d) datasets for (causal) Bayesian networks

to regular time series (Runge et al., 2017), this is not straightforward for datasets involving irregular occurrences of events. In general, one cannot directly deploy standard constraint-based testers from the vast literature on causal networks for the PGEM setting, since here we are interested in testing for process independence among multivariate event processes rather than conditional independence among multiple random variables.

**Contributions.** In this paper, we make the following contributions: 1) we explore process independence in PGEMs and formalize dynamic Markov properties, consistent with prior work on graphical event models; 2) we propose constraint-based algorithms for learning PGEMs, including a max-min parents (MMP) algorithm, as well as two process independence testers. One of these testers estimates the influence of a candidate parent in a Boolean function, whereas the other approximates a likelihood ratio; 3) we conduct an experimental investigation comparing the proposed methods with state-of-the-art approaches. We show some improvements in F1 score for structure discovery using synthetic event datasets generated from PGEMs, as well as an increase in the number of parents identified for select real-world datasets, but more work is required in the future to tackle this difficult task.

## 2. Notation and Background

**Graph Notation.** We review some basic graph related terminology needed for future sections. $\mathcal{G} = (\mathcal{L}, \mathcal{E})$ refers to a directed graph over a set of nodes $\mathcal{L}$ and with directed edges $\mathcal{E}$ represented as ordered pairs from $\mathcal{L} \times \mathcal{L}$. A *path* in $\mathcal{G}$ is a sequence of nodes with edges between successive pairs of nodes, oriented in either direction. A path is *directed* if the sequence only has edges pointing forward in order, and is *trivial* if the sequence has cardinality 1. Node $X$ on a path is a *collider* only if there are directed edges into $X$ from both the nodes before and after it in the sequence; it is referred to as a *non-collider* otherwise. $\mathbf{U}$ refers to the *parents* of a node $X$ in $\mathcal{G}$. *Ancestors* of a node $X$ include $X$ as well as all nodes with a directed path emanating from them to $X$.

We are now armed with the necessary notation required to define a graph separation criterion – *$d^*$-separation*, which is a modification of the well-known *d-separation* for Bayesian networks. This will be used to define an important Markov property later.

**Definition 1** *A path $d^*$-connects nodes $X$ and $Y$ given the set of vertices $\mathbf{Z}$ in graph $\mathcal{G}$ if every collider on the path is an ancestor of $\mathbf{Z}$ and every non-collider is not in $\mathbf{Z}$. For sets $\mathbf{X}, \mathbf{Y}, \mathbf{Z} \subset \mathcal{L}$ s.t. $\mathbf{Y} \cap \mathbf{Z} = \emptyset$, $\mathbf{X}$ is $d^*$-separated from $\mathbf{Y}$ by $\mathbf{Z}$ in $\mathcal{G}$ if and only if there does not exist a non-trivial path that $d^*$-connects any node in $\mathbf{X}$ to any node in $\mathbf{Y}$ given $\mathbf{Z}$ in $\mathcal{G}$.*

**Marked Point Processes and GEMs.** Event datasets involve a single stream or multiple independent streams of events in the form $D = \{(l_i, t_i)\}_{i=1}^N$, where $t_i$ is the occurrence time of the $i^{th}$ event, $t_i \in \mathbb{R}^+$, assumed temporally ordered between start time $t_0 = 0$ and final time $t_{N+1} = T$, and $l_i$ is an event label/type belonging to an alphabet $\mathcal{L}$. We refer to $T$ as the time horizon of the event dataset. These sorts of datasets are widely available across domains; examples include datasets involving web logs, customer transactions, network notifications, political events, financial events, insurance claims, health episodes, etc. Figure 1(a) depicts an example of a single stream event dataset over 3 recurring event labels from the domain of healthcare.

A marked point process for event streams involving event labels from $\mathcal{L}$ is associated with counting processes for each label (Daley and Vere-Jones, 2002). Prior work uses a Doob-Meyer decomposition to show that a conditional intensity function that measures the rate at which an

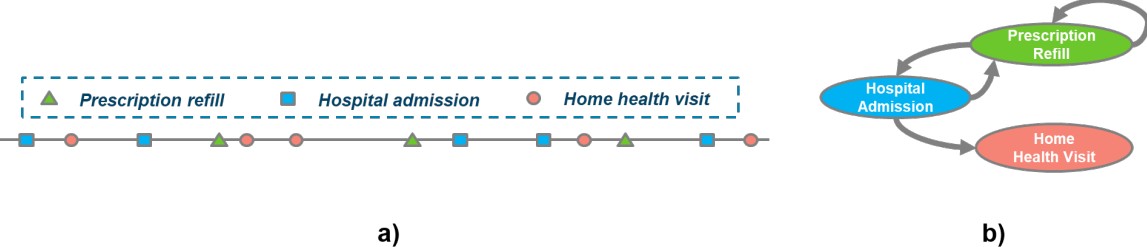

a)                                                                    b)

Figure 1: (a) Illustrative event dataset over 3 event labels from the domain of healthcare: prescription refills, hospital admissions and home health visits. (b) Structure for an illustrative graphical event model (GEM) over the same 3 event labels shown in the event dataset in (a). (Note that both figures are merely illustrative and that the graph in (b) is not related to the dataset in (a).)

event label occurs is sufficient to characterize these processes (Aalen et al., 2008). In general, the conditional intensity for event label $X$ at any time $t$ can be written as a function of the history, i.e. $\lambda_x(t|h_t)$ where $h_t = \{(l_i, t_i) : t_i < t\}$.

Didelez (2008) introduced the notion of process independence among event labels to characterize relationships among the labels' counting processes. The basic idea is that the intensity of one type of event does not depend on certain past events once we know about specific other past events. It should be clear that this is an asymmetric concept, similar to Granger causality. We provide the following informal definition, referring the reader to Didelez (2008) for formal details involving measurability in counting processes:

**Definition 2** *For* $\mathbf{X}, \mathbf{Y}, \mathbf{Z} \subset \mathcal{L}$ *s.t.* $\mathbf{Y} \cap \mathbf{Z} = \emptyset$, $\mathbf{X}$ *is process independent of* $\mathbf{Y}$ *given* $\mathbf{Z}$, *denoted* $\mathbf{Y} \nrightarrow \mathbf{X}|\mathbf{Z}$, *when all event labels in* $\mathbf{X}$ *have conditional intensities such that if the historical occurrences of label set* $\mathbf{Z}$ *are known, then those of label set* $\mathbf{Y}$ *do not provide any further information.*

Graphical event models specify local historical (in)dependencies among the labels' counting processes (Didelez, 2008; Meek, 2014). They can be viewed as representations that indicate how various events labels are generated over time, given the historical occurrences of their parents in the graph. Thus, the conditional intensity for an arbitrary label $X$ at any time $t$ depends only on historical occurrences of its parent event labels, implying that $\lambda_x(t|h_t) = \lambda_x(t|[h(\mathbf{U})]_t)$, where $\mathbf{U}$ are $X$'s parents and $[h(\mathbf{U})]_t$ is the history restricted to labels in set $\mathbf{U}$, i.e. only the historical occurrences of event labels in set $\mathbf{U}$ are considered. Figure 1(b) depicts an illustrative GEM over the same 3 event labels from Figure 1(a). In this example, given the historical occurrences of hospital admissions, the rate of home health visits does not depend on prior occurrences of prescription refills. Note that GEMs as described here may in general involve self loops.

**PGEMs.** A proximal graphical event model is a particular kind of GEM where only the most recent historical occurrences of a node $X$'s parents $\mathbf{U}$ within corresponding time windows affect its conditional intensity (Bhattacharjya et al., 2018). At the structural level, the relationships in a PGEM are therefore described just like any GEM, with a graph $\mathcal{G}$ where there is a node for every event type in $\mathcal{L}$. We continue to use $X$ to refer to an arbitrary node in the PGEM graph and denote $\mathbf{U}$ as its set of parents.

A PGEM also includes quantitative information along with the qualitative structure. Every edge in the graph has an associated time interval (window) from a set $\mathcal{W}$, which specifies the recent

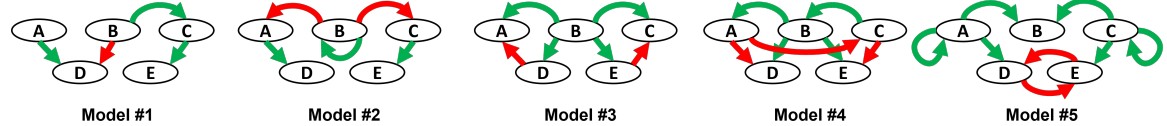

Figure 2: Graphs of 5 example PGEMs, each with 5 nodes, numbered in increasing order of graph density. Green and red arcs indicate excitation and inhibition effects respectively.

time period that the model is sensitive to, with regards to historical dependence for every edge. In addition, a PGEM includes conditional intensity rate parameters $\Lambda = \{\lambda_{x|\mathbf{u}}^{w_x} : \forall X \in \mathcal{L}\}$ where the conditioning is on $\mathbf{u}$, which is an instantiation of $X$'s parents – one of $2^{|\mathbf{U}|}$ possible binary vectors, analogous to a Bayesian network with binary random variables. Conditional intensity parameters represent the rate at which an event label occurs at any time, given the historical condition of its parents, i.e. whether or not a parent has occurred at least once in the corresponding window. The superscript $w_x$ here refers to the set of all windows corresponding to edges that lead into $X$; we omit this in expressions to avoid notational clutter. In a PGEM, the log likelihood for node $X$ given parents $\mathbf{U}$, with windows $w_x$ and conditional intensities $\lambda_{x|\mathbf{u}}$ is:

$$\log L(X|\mathbf{U}) = \sum_{\mathbf{u}} \left( -\lambda_{x|\mathbf{u}} D(\mathbf{u}) + N(x; \mathbf{u}) \ln(\lambda_{x|\mathbf{u}}) \right), \tag{1}$$

where $N(x; \mathbf{u})$ is the number of times that $X$ is observed in the dataset and that the condition $\mathbf{u}$ is true in the relevant preceding windows, and $D(\mathbf{u})$ is the duration from time $0$ to $T$ where $\mathbf{u}$ is true. Formally, $N(x; \mathbf{u}) = \sum_{i=1}^{N} I(l_i = X) I_{\mathbf{u}}^{w_x}(t_i)$ and $D(\mathbf{u}) = \sum_{i=1}^{N+1} \int_{t_{i-1}}^{t_i} I_{\mathbf{u}}^{w_x}(t) dt$, where $I_{\mathbf{u}}^{w_x}(t)$ is an indicator for whether $\mathbf{u}$ is true at time $t$ as a function of the relevant windows $w_x$. $N(x; \mathbf{u})$ and $D(\mathbf{u})$ are summary statistics that can be computed by scanning through an event dataset.

A complete model for a PGEM is denoted $\mathcal{M}$, where $\mathcal{M} = \{\mathcal{G}, \mathcal{W}, \Lambda\}$. Figure 2 illustrates examples of 5 PGEM graphs, each with 5 nodes. The graphical structure captures process independence. For instance, in model #1, the structure captures the fact that the rate at which $D$ occurs at any time depends on whether $A$ and $B$ occur in their respective time windows. In this figure, the colors of the edges indicate whether the effects from $A$ and $B$ are excitatory (green) or inhibitory (red), i.e. increase or decrease the conditional intensity respectively. Further information about the windows and the conditional intensity parameters for these synthetic models are provided in Appendix A. We will use these models later in an experimental investigation.

## 3. Causal Proximal Graphical Event Models

### 3.1. Causality and Process Independence

Causality is naturally related to processes developing over time. Pearl (2009) refers to the word 'mechanism' several times in his seminal work, emphasizing the importance of understanding the inner workings of a system for making causal inferences. We take a mechanistic view of the causal temporal dynamics in event processes (Cox, 1992; Aalen et al., 2012). Specifically, we assume an underlying causal marked point process $\mathcal{P}_{\mathcal{M}}$ associated with a PGEM $\mathcal{M}$ which includes graph $\mathcal{G}$.

For any GEM (including a PGEM), there are certain inherent (local) process independencies that are defined by the construction of a GEM graph $\mathcal{G}$ for an underlying marked point process $\mathcal{P}$.

We follow Didelez (2008) and refer to this as the local dynamic Markov property, where the word 'dynamic' is used to distinguish from the analogous property in (causal) Bayesian networks, which is the local Markov property. In general, we refer the reader to the relevant prior work mentioned in this section to appreciate the analogies with concepts from the Bayesian networks literature.

**Definition 3** *A marked point process $\mathcal{P}$ satisfies the local dynamic Markov property w.r.t graph $\mathcal{G}$ if $\mathcal{L} \setminus \mathbf{U} \not\to X | \mathbf{U} \ \forall X \in \mathcal{L}$.*

**Lemma 4** *A marked point process $\mathcal{P}_\mathcal{M}$ corresponding to a PGEM satisfies the local dynamic Markov property with respect to the PGEM graph $\mathcal{G}$.*

In the work on atemporal causal discovery as modeled by (causal) Bayesian networks, a graphical separation method known as $d$-separation is used to infer additional conditional independencies in the underlying joint probability distribution (Verma and Pearl, 1990; Spirtes et al., 2001). Its modification to allow for self loops, $d^*$-separation, was briefly defined in the previous section. This idea of 'reading off' additional independencies from the graph was extended by Didelez (2008) and subsequently by Meek (2014) to process independence for graphical event models; Meek's extension was to enable an event label to be independent of its own history. We follow this convention, using $\delta^*$-separation as a means of defining the global dynamic Markov property.

**Definition 5** *For $\mathbf{X}, \mathbf{Y}, \mathbf{Z} \subset \mathcal{L}$ s.t. $\mathbf{Y} \cap \mathbf{Z} = \emptyset$, $\mathbf{X}$ is $\delta^*$-separated from $\mathbf{Y}$ by $\mathbf{Z}$ in graph $\mathcal{G}$ if and only if $\mathbf{X}$ is $d^*$-separated from $\mathbf{Y}$ by $\mathbf{Z}$ in the graph formed by deleting any non self-loop outgoing edges from $\mathbf{X}$ in $\mathcal{G}$.*

**Definition 6** *A marked point process $\mathcal{P}$ satisfies the global dynamic Markov property w.r.t graph $\mathcal{G}$ if for labels $X, Y$ and set $\mathbf{Z} \subset \mathcal{L}$ s.t. $Y \cap \mathbf{Z} = \emptyset$, $X$ is $\delta^*$-separated from $Y$ by $\mathbf{Z}$ in $\mathcal{G} \implies Y \not\to X | \mathbf{Z}$.*

The following result confirms that a marked point process associated with a PGEM also satisfies the global dynamic Markov property, in addition to the local dynamic Markov property.

**Theorem 7** *A point process $\mathcal{P}_\mathcal{M}$ corresponding to PGEM $\mathcal{M}$ satisfies the global dynamic Markov property with respect to the PGEM graph $\mathcal{G}$.*

The above result highlights an important implication of $\delta^*$-separation – that one can make additional statements about process independencies from a PGEM graph. For example, consider model #4 in Figure 2. From the local dynamic Markov property applied to event label $E$, $\{A, D\} \not\to E | \{B, C\}$ since $B$ and $C$ are parents of $E$; $A$'s effect on $E$ is indirectly through $C$. From the global dynamic Markov property, we can also say, for instance, that $D \not\to E | \{A, B\}$ because all paths from $D$ to $E$ go through either $A$ or $B$. Note that process independence can be asymmetric in general, although in this instance one can see that $E \not\to D | \{A, B\}$ by applying the local dynamic Markov property to $D$.

In (causal) Bayesian networks, a family of algorithms known as constraint-based methods recover the underlying structure by estimating from the data whether certain conditional independencies between the variables hold; the PC algorithm is a classic example (Spirtes et al., 2001). We apply this approach to GEMs, estimating process independencies between event labels. Algorithm 1 outlines the PC algorithm as applied to GEMs. Since there are no acyclicity constraints in GEMs,

one can learn the parents for each target node $X \in \mathcal{L}$ separately. The algorithm works by growing the conditioning set $\mathbf{Z}$ until process independence is discovered, in which case the edge from the candidate parent $Y$ to target node $X$ is removed. Note that a process independence tester needs to be plugged into the algorithm, which we assume outputs a score that is monotonically increasing in the amount of dependence. Independence is assumed when the tester's score is less than a function $g(\cdot)$ of a threshold $\alpha$. We introduce two such testers for PGEMs in the next sub-section.

Together with the global dynamic Markov property, the following assumption, which is analogous to the causal faithfulness assumption in Bayesian networks, helps specify processes where the only dependencies are those that can be determined by $\delta^*$-separation. For such processes, and when a perfect process independence tester is available, the PC algorithm for PGEMs is sound and complete, i.e. the true graph is learned.

---

**Algorithm 1** PC Algorithm for Structure Discovery in GEMs

---

**Data:** Event label $X \in \mathcal{L}$, event dataset $D$ (over $\mathcal{L}$), threshold parameter for tester $\alpha$

**Result:** Parents $\mathbf{U}$ for $X$

$\mathbf{U} = \mathcal{L}$

**for** *all $Y$ in $\mathcal{L}$* **do**
   flag = False, $n = 0$, $\mathbf{Z}^* = \mathbf{U} \setminus Y$
   **while** $n \leq |\mathbf{Z}^*|$ *and flag = False* **do**
      **for** *all $\mathbf{Z}$ that are subsets of size $n$ in* $\mathbf{Z}^*$ **do**
         Obtain score from a process independence test
         **if** *score $\leq \tau = g(\alpha)$ (indicating process independence)* **then**
            flag = True, $\mathbf{U} = \mathbf{U} \setminus Y$
            Break from loop
         **end**
      **end**
      $n = n + 1$
   **end**
**end**

---

**Algorithm 2** MMP (Max-Min Parents) Algorithm for Structure Discovery in GEMs

---

**Data:** Event label $X \in \mathcal{L}$, event dataset $D$ (over $\mathcal{L}$), threshold parameters for tester $\alpha$ and $\beta$

**Result:** Parents $\mathbf{U}$ for $X$

$\mathbf{U} = \emptyset$

**Phase I:**

$t = 1$

**while** $t \neq 0$ **do**
   $P = \mathrm{argmaxmin}_\alpha(X; \mathbf{U})$
   $t = \mathrm{maxmin}_\alpha(X; \mathbf{U})$
   **if** $t > 0$ **then**
      $\mathbf{U} = \mathbf{U} \cup P$
   **end**
**end**

**Phase II:**

**for** $Y \in \mathbf{U}$ **do**
   $t = \min_{F \subset \mathbf{X} \setminus Y} \mathrm{Assoc}_\beta(Y \to X; F)$
   **if** $t = 0$ **then**
      $\mathbf{U} = \mathbf{U} \setminus Y$
   **end**
**end**

---

**Definition 8** *A marked point process $\mathcal{P}$ satisfies the causal dependence assumption w.r.t graph $\mathcal{G}$ if for sets $\mathbf{X}, \mathbf{Y}, \mathbf{Z} \subset \mathcal{L}$ s.t. $\mathbf{Y} \cap \mathbf{Z} = \emptyset$, $\mathbf{Y} \not\to \mathbf{X}|\mathbf{Z} \implies \mathbf{X}$ is $\delta^*$-separated from $\mathbf{Y}$ given $\mathbf{Z}$ in $\mathcal{G}$.*

Note that discovering the true structure is only possible even under strict assumptions when the parameters can be estimated perfectly from data; this is often violated in practice when only a finite amount of data is available. Existing literature has studied the impact of faithfulness violation for i.i.d. data (Robins et al., 2003; Uhler et al., 2013).

**Theorem 9** *If a marked point process $\mathcal{P}_\mathcal{M}$ corresponding to a PGEM satisfies the causal dependence assumption, the PC algorithm with a perfect process independence tester recovers the true underlying PGEM graph $\mathcal{G}$.*

Inspired by the max-min hill climbing algorithm for Bayesian networks (Tsamardinos et al., 2006), we propose a variant for GEMs structure learning called the max-min parents (MMP) algorithm, outlined in Algorithm 2. It consists of two phases: the first phase picks candidate parents while the second prunes the list picked in the first phase. Again, we assume that a tester returns a score which measures dependence. We define a measure of association from the score and a threshold $\alpha$:

$$\text{Assoc}_\alpha(Y \to X; \mathbf{Z}) = \max(\alpha, (score)) - \alpha \tag{2}$$

We also define the functions: $\text{maxmin}_\alpha(X; \mathbf{Z}) = \max_{X \neq Y} \min_{\mathbf{F} \subset \mathbf{Z}} \text{Assoc}_\alpha(Y \to X; \mathbf{F})$ and $\text{argmaxmin}_\alpha(X; \mathbf{Z}) = \arg\max_{Y \neq X} \min_{\mathbf{F} \subset \mathbf{Z}} \text{Assoc}_\alpha(Y \to X; \mathbf{F})$.

The algorithm proceeds as follows: given a current set of parents in Phase 1, conditioned on all subsets of current parent set, we check for the minimum association measure for a candidate parent. Amongst several choices, we pick the candidate parent with the maximum minimum measure of association. If all parents are picked and if the tester is accurate (outputs a score $\leq \alpha$ on independence), this measure will be zero. If a parent is left out, the measure will always be non-zero. Phase 1 may end with including extraneous nodes other than the true parents. Phase 2 attempts to eliminate these extra nodes by conditioning on all subsets of the remaining parent set to eliminate. Again, extraneous nodes will be thrown for a perfect process independence tester.

MMP has some potential advantages over the PC algorithm, which starts from a complete graph and relies only on process independence relations to reduce the number of edges. Statistically, with a larger conditioning set, due to noise it can be more difficult for PC to detect independence. In contrast, the MMP algorithm first builds a candidate parent set which is then pruned. The MMP algorithm's first phase is not affected if process independence is not detected properly while conditional dependence needs to be picked. It may only lead to a larger candidate parent set. However, in Phase 2, thresholds can be made stricter to strengthen conditional independencies to prune the set obtained at the end of Phase 1. This could reduce false positive rates better than the PC algorithm.

### 3.2. Process Independence Testers

The main challenge in learning with the PC and MMP algorithms lies in finding an effective tester. We propose two process independence testers that use properties of the PGEM representation. As far as we are aware, there is not much prior work on testers for models within the broad family of GEMs. Both our testers take as input a target node $X$, a conditioning set of event labels $\mathbf{Z}$, a candidate parent $Y \notin \mathbf{Z}$, and of course an event dataset $\mathcal{D}$, returning a score as output. In the PC algorithm, if the score is less than a specified threshold, process independence is declared, i.e. $Y \not\to X | \mathbf{Z}$. The MMP algorithm is similar except that it has two thresholds – one for the forward phase as the graph grows, and one for the backward phase where spurious edges are removed. For experiments, these two thresholds are set to be identical.

For both testers, we assume that when $X$ has parents $\mathbf{U}$, the windows for edges into $X$ ($w_x$) are known, and therefore one can easily compute the conditional intensity parameters through maximum likelihood estimation using summary statistics, $\hat{\lambda}_{x|\mathbf{u}} = \frac{N(x;\mathbf{u})}{D(\mathbf{u})}$ (see equation 1). Specifically,

we use the 'independent windows' approach from Bhattacharjya et al. (2018) to estimate windows, where the window for each edge $Y$ to $X$ is estimated by assuming that $X$ has no other parent(s). Finding optimal windows, i.e. those that maximize the log likelihood, is known to be a hard combinatorial problem, so it is necessary to make an approximation of this sort for tractability.

### 3.2.1. NORMALIZED INFLUENCE (NI) TESTER

For a PGEM, the conditional intensities $\lambda_{x|\mathbf{u}}$ for a node $X$ with parents $\mathbf{U} = \{Y, \mathbf{Z}\}$ are Boolean functions from $\{0, 1\}^{|\mathbf{Z}|+1} \to \mathbb{R}$, therefore one can estimate the *influence* (or synonymously *sensitivity*) of $Y$ on the function $\lambda_{x|\mathbf{u}}$ through analysis of this Boolean function (O'Donnell, 2014). For a multi-dimensional Boolean function $f(\cdot)$, the influence of a variable $i$ is defined as $E[(D_i f)^2]$ where $D_i f$ is the derivative operator measuring the change in the function from toggling the $i^{th}$ variable bit from 0 to 1. Formally, $D_i f(\theta) = \frac{f(\theta^{(i\to 1)}) - f(\theta^{(i\to 0)})}{2}$. We normalize this influence by the second moment of the function, $E[f^2]$, to gauge $Y$'s contribution to the variance. Computing $\frac{E[(D_i f)^2]}{E[f^2]}$ for a PGEM conditional intensity function $f = \lambda_{x|\mathbf{u}}$ with $i = Y$ and $\mathbf{U} = \{Y, \mathbf{Z}\}$ results in the following normalized influence score:

$$\text{NI score} = \frac{1}{2} \frac{\sum_{\mathbf{z}} \left(\lambda_{x|y,\mathbf{z}} - \lambda_{x|\bar{y},\mathbf{z}}\right)^2}{\sum_{\mathbf{z}} \left(\lambda_{x|y,\mathbf{z}} + \lambda_{x|\bar{y},\mathbf{z}}\right)^2}, \tag{3}$$

where $y$ and $\bar{y}$ in the subscript for conditional intensity indicate the parental states where $Y$ has and has not occurred in its window, respectively. The NI score estimates the contribution of $Y$ to a Boolean function that also includes event labels $\mathbf{Z}$. If the score is less than a threshold $\tau$, we declare that $Y$ does not have enough additional impact on $X$ given $\mathbf{Z}$. For this tester, we set the threshold $\tau = \frac{\alpha}{|\mathbf{Z}|+1}$ for some threshold parameter $\alpha$, so as to adjust the level of meaningful contribution depending on the size of $\mathbf{Z}$. For instance, if $\alpha = 0.1$ and $|\mathbf{Z}| = 3$, then a score $\leq 2.5\%$ implies process independence, i.e. $Y \not\to X | \mathbf{Z}$.

### 3.2.2. LIKELIHOOD RATIO (LR) TESTER

For this tester, we consider 2 models – a coarser model where the set of parents for a node $X$ is $\mathbf{Z}$, and a more refined model where $Y$ is also a parent in addition. Note that these are nested models for a PGEM, in the sense that the conditional intensity parameters $\lambda_{x|\mathbf{u}}$ with parents $\{Y, \mathbf{Z}\}$ can subsume the case where $Y$ is not a parent by setting $\lambda_{x|y,\mathbf{z}} = \lambda_{x|\bar{y},\mathbf{z}} \forall \mathbf{z}$.

A function of the ratio between two likelihoods is often used to compare models in hypothesis testing. Specifically, the ratio compares a likelihood found by maximizing over a broader class of models and another found after imposing some constraints. Adapting this to PGEMs, we use $\text{LR} = -2 \left[\log L^*(X|Y, \mathbf{Z}) - \log L^*(X|\mathbf{Z})\right]$, where the maximum log likelihoods $L^*(X|\mathbf{U})$ are found by replacing maximum likelihood estimates for conditional intensities, $\hat{\lambda}_{x|\mathbf{u}} = \frac{N(x;\mathbf{u})}{D(\mathbf{u})}$, in equation 1.

A classic result states that for a class of nested models, the LR statistic asymptotically tends to a chi-squared distribution with number of degrees equal to the difference between the number of parameters of the two nested models (Wilks, 1938). For PGEMs, this difference is $2^{|\mathbf{Z}|}$; we therefore use the following score to test for process independence:

$$\text{LR score} = F_{\chi^2_{2|\mathbf{Z}|}} \left(-2 \left[\log L^*(X|Y, \mathbf{Z}) - \log L^*(X|\mathbf{Z})\right]\right), \tag{4}$$

where $F(\cdot)$ is the cumulative distribution function of a chi-squared random variable with $2^{|\mathbf{Z}|}$ degrees of freedom. We declare process independence $Y \not\rightarrow X|\mathbf{Z}$ when the score is less than a threshold $\tau$, to determine that the gain in the log likelihood from including $Y$ with $\mathbf{Z}$ is close to $0$ and therefore not substantial. For this tester, we set the threshold $\tau = \alpha$ for threshold parameter $\alpha$ since the score is a probability. Please note that it is merely $1$ minus the p-value of the test statistic. This deliberate non standard choice is chosen just to have a score that is high when dependence is high.

When applied to PGEMs, the LR test statistic and associated score is an approximation as: 1) we are concerned with testers for limited data, 2) the linearity assumptions required of the parameter space for the asymptotic results are not satisfied by the PGEM conditional intensity functions, and 3) marginal models of an underlying PGEM are not PGEMs.

## 4. Experiments

### 4.1. Synthetic Datasets: Structure Discovery for 5 Specific PGEMs

**Setup.** We begin our experimental investigation by considering 5 specific PGEMs, each with 5 nodes. Their graphs are shown in Figure 2 in increasing order of graph density (number of arcs) and therefore complexity. For each model, we generate 20 event streams up to $T = 1000$ days (approximately 3 years in units of days). The complete model specifications, i.e. including windows and conditional intensity parameters for these models are specified in Appendix A.

We compare the proposed testers, as deployed by the PC and MMP algorithms, along with three baselines with the following hyper-parameter settings:

- The score-based BIC learner for PGEMs was run with window increment $\epsilon = 0.001$ for window search, as proposed in Bhattacharjya et al. (2018).

- For the CPCIM learner, which is an algorithm to learn GEMs with piecewise constant conditional intensities over historical basis functions (Parikh et al., 2012), we took the following approach for hyper-parameters. The structural prior $\kappa$ was set to $0.1$. For conjugate prior pseudo-count $\alpha$ and pseudo-duration $\beta$ for each label, we used identical values for all labels. We compute ratio $\rho$ of the total number of all arrivals over all labels to the total duration for all labels (the product of the number of labels and the horizon $T$ under consideration) which provides an empirically based estimate of the arrival rate. We ran experiments using $\alpha = K\rho$, $\beta = K$, for various values of $K = 10, 20, \ldots$, where higher values of $K$ increase the prior's influence. $K = 20$ was chosen. Intervals of the form $[t - t^*, t)$ are the basis functions, where we chose $t^* \in \{1, 2, 3, 4, 5, 6, 7, 15, 30, 45, 60, 75, 90, 180\}$.

- We also considered CAUSE, which fits a neural point process for an event stream and then extracts a graph structure from a Granger causality statistic based on an axiomatic attribution method (Zhang et al., 2020). This is a relatively recent approach that is supposed to be appropriate for GEMs in general. We use publicly available code[1] at the default settings.

We determine the optimal threshold $\tau$ for the testers by searching over a grid of threshold parameters (treated as hyper-parameters) on a training set of the first 10 event streams for each model. Each learner is then evaluated with its optimal threshold setting on a test set of the remaining 10

---

1. https://github.com/razhangwei/CAUSE

Table 1: Comparing structure discovery F1 scores over 5 example PGEMs.

| Model | CPCIM | CAUSE | BIC | PC-LR | PC-NI | MMP-LR | MMP-NI |
|-------|-------|-------|-----|-------|-------|--------|--------|
| #1 | $0 \pm 0$ | $0.27 \pm 0$ | $0.38 \pm 0.3$ | $0.3 \pm 0.07$ | $0.29 \pm 0.04$ | $\mathbf{0.49} \pm 0.22$ | $0.29 \pm 0.04$ |
| #2 | $0 \pm 0$ | $0.33 \pm 0$ | $0.16 \pm 0.16$ | $0.31 \pm 0.14$ | $\mathbf{0.4} \pm 0.14$ | $0.36 \pm 0.06$ | $0.3 \pm 0.2$ |
| #3 | $0 \pm 0$ | $\mathbf{0.39} \pm 0$ | $0.29 \pm 0.13$ | $0.37 \pm 0.1$ | $\mathbf{0.39} \pm 0.1$ | $\mathbf{0.39} \pm 0.11$ | $\mathbf{0.39} \pm 0.1$ |
| #4 | $0.25 \pm 0$ | $\mathbf{0.44} \pm 0$ | $0.23 \pm 0.22$ | $0.3 \pm 0.1$ | $0.41 \pm 0.06$ | $0.39 \pm 0.1$ | $0.41 \pm 0.06$ |
| #5 | $0.22 \pm 0$ | $0.48 \pm 0$ | $0.15 \pm 0.2$ | $0.41 \pm 0.1$ | $\mathbf{0.5} \pm 0.12$ | $0.49 \pm 0.16$ | $\mathbf{0.5} \pm 0.12$ |

event streams per model. Tester threshold parameters for training were chosen from a grid $\mathcal{T}$ as follows: $\{0.3, 0.25, 0.2, 0.15, 0.1, 0.05, 0.03\}$ for the NI tester and $\{0.99, 0.95, 0.9, 0.8, 0.7, 0.6, 0.5\}$ for the LR tester.

**Results.** Table 1 shows the mean F1 scores along with the error, as measured by half of the 90th and 10th percentiles across the 10 event streams, for each model-learner combination. CPCIM often cannot recover any true parents. BIC exhibits poor F1 scores as it learns sparse graphs, usually with good precision but poor recall. While BIC performs well in comparison to the PC learners for the sparsest model (model #1) due to fewer arcs to recall, performance deteriorates for the more complex models. In contrast, the strong baseline CAUSE generally performs well for the more dense graphs but poorly for the sparse ones. The PC and MMP algorithms are generally similar in their performance. The NI tester works better than the LR tester for the more complex models. MMP-LR is robust in that it performs well across the 5 PGEMs, and appears to be a reasonable choice for PGEM structure discovery. MMP has the advantage of being more efficient than PC.

Note that the NI tester leverages the proximal assumption in the graphical event models that were studied, whereas the LR tester is more general and could potentially be more robust over a broader class of models, although that remains to be empirically demonstrated. The results however generally reveal the testers to be comparable. Note that F1 scores are generally low throughout, exhibiting that the task of uncovering causal relations from event streams is a challenging one and requires further investigation.

### 4.2. Synthetic Datasets: Structure Discovery for 20 Randomly Generated PGEMs

**Setup.** To illustrate the generality of our observation that the newly proposed constraint-based methods (PC and MMP) improve recall and therefore the F1 score beyond the score-based BIC learner, we repeat the previous experiment with 20 randomly generated PGEMs.

PGEMs were randomly generated similar to the approach described in the supplementary material in Bhattacharjya et al. (2018). For a PGEM over label set $\mathcal{L}$, for each node, the number of its parents $K$ are chosen uniformly from the parameters $K_{min} = 0, \cdots, K_{max} = \lfloor |\mathcal{L}|/2 \rfloor$ in integer increments. A random subset of size $K$ from $\mathcal{L}$ is then chosen as its parent set. Windows for each edge are generated uniformly from $w_{min} = 15$ to $w_{max} = 30$ in increments of $\Delta w = 5$. For the conditional intensity rates, we assume that each node's parent either has a multiplicative amplification or damping rate beyond a baseline rate of $r/|\mathcal{L}|$, where $r$ is generated uniformly between $r_{min} = 0.05$ and $r_{max} = 0.2$. Nodes that always increase occurrence rate for their children are obtained by randomly choosing a subset $\mathcal{L}_A$ of size $K_A = \lfloor |\mathcal{L}|/2 \rfloor$ from $\mathcal{L}$. Nodes in the sets $\mathcal{L}_A$ and $\mathcal{L} \backslash \mathcal{L}_A$ have an amplification and damping rate of $\gamma_A = 1.5$ and $\gamma_D = 0.25$ respectively. These numbers are chosen to roughly keep the number of events $N$ generated by each model to be

Table 2: Comparing structure discovery F1 scores for two different dataset sizes, with results averaged over 20 randomly generated PGEMs.

| $T$ (End Time) | BIC | PC-LR | PC-NI | MMP-LR | MMP-NI |
|---|---|---|---|---|---|
| 500 | $0.28 \pm 0.29$ | $0.31 \pm 0.17$ | $\mathbf{0.33} \pm 0.11$ | $\mathbf{0.33} \pm 0.17$ | $\mathbf{0.33} \pm 0.17$ |
| 1000 | $0.3 \pm 0.21$ | $\mathbf{0.39} \pm 0.16$ | $0.37 \pm 0.14$ | $0.37 \pm 0.18$ | $0.37 \pm 0.14$ |

commensurate with $T$, but this is not enforced rigorously, thereby allowing the dataset sizes to vary across models.

The experimental setup is otherwise identical to earlier, except that the results are averaged over the 20 models as well as the 10 event streams over each model. This analysis is done for synthetic datasets of varying lengths: $T = 500$ and $T = 1000$, to appreciate the effect of data size.

**Results.** We observe from Table 2 that PC and MMP perform similarly and that again they beat the BIC score approach. All methods exhibit improved performance with more data.

### 4.3. Real Datasets: Number of Parents

**Setup.** Here we consider a select few real event datasets from various domains with unknown ground truth graphs. We investigate how many additional parents could potentially be learned by a constraint-based method. We stress that this does not mean that the proposed methods only recover more true parents. In fact, our synthetic experiments reveal that the tester-based learners almost always involve false positives. The intent is merely to show how under reasonable values of thresholds, one can recover more than the sparser graphs learned by BIC. This could be useful in practice for analysts and scientists, particularly when combined with their domain knowledge.

The datasets considered are: 1) the books Leviathan and the Bible, available from the SPMF data mining library (Fournier-Viger et al., 2014). The 100 most frequent words were removed and the next most frequent $M = 10$ words were used as labels and their positions in the books as time stamps; 2) political events in Argentina, Mexico and Venezuela, three countries from the curated version of the ICEWS political event dataset (Bhattacharjya et al., 2018); 3) MIMIC – a medical dataset with clinical visit records by patients; this has been used in other event modeling work (Du et al., 2016).

**Results.** Table 3 compares the average number of parents per node that are learned by the baseline BIC and MMP-LR. We use $\{0.99, 0.95, 0.9\}$ as the identical threshold parameters for MMP-LR. Recall that a higher valued threshold results in more process independencies and therefore a smaller parent set. Only the MMP is considered here as it is more efficient than PC and can handle the larger datasets. Thresholds are set to be identical, i.e. $\alpha = \beta$. We observe that MMP-LR learns between twice and four times the number of parents than BIC for these datasets from varying domains. MMP-NI learns graphs that are even denser than MMP-LR but the numbers are not shown here.

## 5. Conclusions

In this paper, we presented one of the first constraint-based investigations of graphical event models, including a novel max-min parents algorithm and two process independence testers for an important family of models – proximal graphical event models. Learning causal graphical representations

Table 3: Comparing average number of parents across learners over select real datasets.

| Dataset | BIC | MMP-LR | | |
|---------|-----|--------|------|------|
| | | 0.99 | 0.95 | 0.9 |
| Leviathan | 1 | 3.2 | 3.7 | 3.9 |
| Bible | 1.5 | 4.3 | 4.7 | 4.9 |
| Argentina | 1.12 | 2.26 | 2.8 | 2.91 |
| Mexico | 1.04 | 1.87 | 2.19 | 2.3 |
| Venezuela | 0.91 | 2.01 | 2.36 | 2.54 |
| MIMIC | 0.49 | 0.96 | 1.11 | 1.17 |

for event streams is challenging, and although our proposed approaches show improvement over baselines for the task of structure discovery in PGEMs, the low F1 scores indicate that there is substantial scope for further advances. Indeed, there is scope for future work around testers that could potentially detect process independence across a reasonably broad class of models within the GEMs family. The major difficulty is that the complex nature of such data results in time dependent confounding where the temporal interaction between occurrences makes it difficult to extricate direct causes. Keiding (1999) provides some discussion about the challenges in this regard. We anticipate further advances in the future around the challenging area of structure discovery for stochastic processes in general.

## Acknowledgments

We thank the reviewers for helpful feedback and suggestions.

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

## Appendix A. Model Details for PGEMs for Synthetic Data Experiments

We provide details about the 5 PGEMs in Figure 2. In what follows, the windows corresponding to the parents are listed in the same order as parents. We use binary vectors to indicate parental states, again in the same order as listed parents. For instance, if a node $A$ has parents $[B, C]$ then windows $[15, 30]$ represent information that the windows from $B$ and $C$ to $A$ respectively are 15 and 30. The binary parental state $[0, 1]$ implies that only $C$ has occurred in its window, whereas $[1, 1]$ represents the case where both $B$ and $C$ have occurred in their respective windows.

**Model** $\#1$

- parents = {'A': [], 'B': [], 'C': ['B'], 'D': ['A', 'B'], 'E': [C']}

- windows = {'A': [], 'B': [], 'C': [15], 'D': [15, 30], 'E': [15]}

- lambdas = { 'A': {[]: 0.2}, 'B': {[]: 0.05}, 'C': {[0]: 0.2, [1]: 0.3}, 'D': {[0, 0]: 0.1, [0, 1]: 0.05, [1, 0]: 0.3, [1, 1]: 0.2}, 'E': {[0]: 0.1, [1]: 0.3}, }

**Model** $\#2$

- parents = {'A': ['B'], 'B': ['B'], 'C': ['B'], 'D': ['A'], 'E': ['C']}

- windows = {'A': [15], 'B': [30], 'C': [15], 'D': [30], 'E': [30]}

- lambdas = { 'A': {[0]: 0.3, [1]: 0.2}, 'B': {[0]: 0.2, [1]: 0.4}, 'C': {[0]: 0.4, [1]: 0.1}, 'D': {[0]: 0.05, [1]: 0.2}, 'E': {[0]: 0.1, [1]: 0.3} }

**Model** $\#3$

- parents = {'A': ['B', 'D'], 'B': [], 'C': ['B', 'E'], 'D': ['B'], 'E': ['B']}

- windows = {'A': [15, 30], 'B': [], 'C': [15, 30], 'D': [30], 'E': [30]}

- lambdas = { 'A': {[0,0]: 0.1, [0,1]: 0.05, [1,0]: 0.3, [1,1]: 0.2}, 'B': {[]: 0.2}, 'C': {[0,0]: 0.2, [0,1]: 0.05, [1,0]: 0.4, [1,1]: 0.3}, 'D': {[0]: 0.1, [1]: 0.2}, 'E': {[0]: 0.1, [1]: 0.4} }

**Model** #4

- parents = {'A': ['B'], 'B': ['C'], 'C': ['A'], 'D': ['A', 'B'], 'E': ['B', 'C']}

- windows = {'A': [15], 'B': [30], 'C': [15], 'D': [15, 30], 'E': [30, 15]}

- lambdas = { 'A': {[0]: 0.05, [1]: 0.2}, 'B': {[0]: 0.1, [1]: 0.3}, 'C': {[0]: 0.4, [1]: 0.2}, 'D': {[0, 0]: 0.1, [0, 1]: 0.3, [1, 0]: 0.05, [1, 1]: 0.2}, 'E': {[0, 0]: 0.1, [0, 1]: 0.02, [1, 0]: 0.4, [1, 1]: 0.1} }

**Model** #5

- parents = {'A': ['A'], 'B': ['A', 'C'], 'C': ['C'], 'D': ['A', 'E'], 'E': ['C', 'D']}

- windows = {'A': [15], 'B': [30, 30], 'C': [15], 'D': [15, 30], 'E': [15, 30]}

- lambdas = { 'A': {[0]: 0.1, [1]: 0.3}, 'B': {[0,0]: 0.01, [0,1]: 0.05, [1,0]: 0.1, [1,1]: 0.5}, 'C': {[0]: 0.2, [1]: 0.4}, 'D': {[0, 0]: 0.05, [0, 1]: 0.02, [1, 0]: 0.2, [1, 1]: 0.1}, 'E': {[0, 0]: 0.1, [0, 1]: 0.01, [1, 0]: 0.3, [1, 1]: 0.1}, }

## Appendix B. Proofs

### B.1. Proof for Lemma 4

This follows from the definition of a GEM. For a PGEM, if a node $X$ has parent nodes $\mathbf{U}$, then at any time with parent condition $\mathbf{u}$ (as determined by recent historical occurrences in the corresponding windows), the rate at which $X$ occurs is $\lambda_{x|\mathbf{u}}$. Additional information about historical occurrences of any non-parent has no effect on the conditional intensity rate at any time, therefore process independence is true for any non-parent given the history of parent event labels.

### B.2. Proof for Theorem 7

The global Markov property is satisfied when any $\delta^*$-separation statement for valid $X$, $Y$, $\mathbf{Z}$ implies process independence $Y \not\to X|\mathbf{Z}$. The separation itself is defined based on a graph that cuts outgoing edges, except self-loops, from $X$. We refer to this graph as $\tilde{G}_X^D$; the superscript indicates the graph is directed. Consider an undirected graph formed from $\tilde{G}_X^D$ by taking the subgraph over ancestors of $X$, $Y$, $\mathbf{Z}$, and connecting edges between any parents with common children if they are not already connected (this operation is known as 'moralizing'). We denote this as $\tilde{G}_X^U$.

Suppose $X$ is $\delta^*$-separated from $Y$ given $\mathbf{Z}$ in $\tilde{G}_X^D$. We consider a node to be a blocker in a path if it prevents a path from connecting $X$ and $Y$ given $\mathbf{Z}$ for the separation criterion under consideration. Note that if a path is being blocked by a non-collider in $\tilde{G}_X^D$, it will also be blocked by that non-collider in the undirected version $\tilde{G}_X^U$. Consider a path that is blocked by a collider in $\tilde{G}_X^D$. In this path, $X$ must have an incoming edge in $\tilde{G}_X^D$ as outgoing edges have been removed. Furthermore, the collider must not be an ancestor of $\mathbf{Z}$ as it is a path blocker. There must be a path in $\tilde{G}_X^U$ from the corresponding collider node to either $X$ or $Y$. It can be shown that a violation occurs for the assumption of $X$ being $\delta^*$-separated from $Y$ given $\mathbf{Z}$; some other non-blocking path must exist between $X$ and $Y$ as this path cannot include $\mathbf{Z}$, otherwise the collider would be an ancestor of $\mathbf{Z}$, which is not possible. The original path must be blocked by the collider in $\tilde{G}_X^U$. The result follows from applying Theorem 3.4 in Didelez (2008) which uses graphical separation in $\tilde{G}_X^U$.

### B.3. Proof for Theorem 9

The PC algorithm for GEMs is a variation on PC for Bayesian networks with the additional point of simplification that a step for orienting edges to adhere to acyclicity constraints is not needed. Note that the global dynamic Markov property applies to a PGEM from a prior theorem. Together with the causal dependence assumption, this implies that the independencies in the underlying marked point process are the same as those that can be determined from $\delta^*$-separation in the graph. The argument that a perfect process independence tester with the GEMs PC algorithm results in sound and consistent learning follows the argument for the PC algorithm for Bayesian networks. The equivalence of independencies ensures that PC produces no false positives, while the causal dependence assumption ensures no false negatives.

