# OpenReview forum: "Process Independence Testing in Proximal Graphical Event Models"
_cclear.cc/CLeaR/2022/Conference — CLeaR 2022 Poster_

### Official Review · Reviewer_mCqK · 2021-11-19

**Confidence:** 2
**Overall Score:** 7

**Main Review:**

I do not know much about causal discovery or event process analysis, so I cannot speak for the accuracy of technical details in this paper. But as a reader who has very little background knowledge, I found this paper very well written. The exposition is crisp, precise, concise while still providing right amount of information for unfamiliar readers to understand the high level messages of this paper. The idea appears novel and the execution looks very solid and complete. Therefore, I recommend acceptance but due to my limited knowledge my assessment may not be particularly accurate.

**Summary:**

This paper formalizes causal graphs for event process and proposes novel algorithms to discover such causal graphs.

---

> ### Author Response · Authors · 2021-12-03
> **Response**
>
> We thank the reviewer for their positive feedback, particularly around the clarity of writing. We hope to improve further in that regard by addressing comments from other reviewers.

---

### Official Review · Reviewer_d9fn · 2021-11-23

**Confidence:** 3
**Overall Score:** 6

**Main Review:**

The paper introduces the notion of PGEMs, and subsequently it defines their respective notions of separation, Markov conditions and faithfulness. It uses these notions to modify the PC algorithm and introduce an additional algorithm for learning the graph. The empirical evaluation is sufficiently extensive for this study and seems encouraging.

Unfortunately, this is only a borderline paper due to the lack of clarity in the writing. There are several places in which this can be substantially improved. Here are some pointers:

1. Insufficient intuitive comparison is made between causal graphs and PGEMs, and how their notions of separation, Markov conditions, and faithfulness are related.

2. Tying into the previous point, there are many definitions introduced without any intuition about what they represent. For example, it would be useful to to explain how Definition 3 is related to the Causal Markov property.

3. In Figure 1, it is unclear why part b) follows from the timeline presented in part a).

4. The authors should rigorously define the marked point process.

5. The authors should explain in more detail the meaning of a conditional density. Also, "emanate" may not be the best verb to use when describing w_x in the same paragraph.

6. Immediately before the start of section 3.2.1, the meaning of "finding optimal windows" is unclear.

7. In equation 3, the authors have not defined \bar{y}.





**Summary:**

This paper provides the first constraint-based approach for discovery of PGEMs.

---

> ### Author Response · Authors · 2021-12-03
> **Response**
>
> We thank the reviewer for helpful feedback, particularly around how we can further clarify many aspects of the work. Responses to specific points follow:
>
> 1. Some of the connections between Bayesian networks/causal networks (for i.i.d. data) and graphical event models (for event stream data) have been made in prior work (e.g. Didelez 2008). We will make minor edits and provide some guidance to the reader in this regard by addressing point 2 below.
> 2. Many of these definitions are analogs of the corresponding causal/Bayesian network related ideas. For instance, Def. 3 is an analog to the local Markov property where process independence is considered rather than conditional independence. We will clarify this in our exposition.
> 3. Part b) in Fig. 1 does not actually follow from what is shown in a); they are just shown together for convenience and for saving space. They are both merely illustrative. We will clarify this in the caption.
> 4. Given the space limitations, it is tricky for us to formalize marked point processes completely here (from a measure theoretic perspective). Also, we think it could be distracting to the general reader. We will however try to add a few more details.
> 5. Yes, we agree, both about the benefits of having more detail about conditional intensities and use of the word "emanating" as mentioned. We will edit accordingly.
> 6. By optimal windows, we mean those that maximize the log likelihood on the data.
> 7. We thank the reviewer for pointing this out. \bar{y} is used to denote when the event label Y does not occur in the recent history. We will clarify this notation here and possibly also earlier in Section 2.

---

### Official Review · Reviewer_MFoM · 2021-11-25

**Confidence:** 3
**Overall Score:** 6

**Main Review:**

The paper studies proximal graph event models and in particular how to
evaluate process independencies. Such evaluation tests are fundamental
for learning causal structure.

The study of process independencies presents unique challenges on top
of usual causal discovery. While it is already hard to test different
conditional independence requirements among multiple variables,
studying causal structure learning requires testing conditional
independences among different processes to start with---each process
can be high-dimensional object---even if we are just testing the
conditional independence structure among three variables.

The paper makes an ambitious attempt to address this problem, for
which it proposes conditional process independence tests for causal
PGEMs and deploying them in constraint-based structure discovery
algorithms. The key innovation is the treatment of multivariate
conditioning in independence testing, for which it proposes a
normalized influence tester and a likelihood ratio tester. The paper
demonstrates the improved empirical performance in empirical studies.

I think the paper would benefit from the a further explanation with
intuition of why the proposed algorithm is expected to do well. While
the proposals make sense, a constant question I had while reading the
paper is that: multivariate conditioning is an intrinsically hard
problem, so what exactly does the proposal do to solve it? Why or why
not influence is the right metric for such tasks? How should we expect
influence tester to compare with likelihood ratio tester? Are the two
expected to perform similarly or differently in different settings?
When do we expect each of them to perform well? I think answering
these questions would make the paper much stronger.


**Summary:**

review

---

> ### Author Response · Authors · 2021-12-03
> **Response**
>
> We thank the reviewer for helpful feedback. Here are some responses to two major questions:
>
> *Regarding "intuition of why the proposed algorithm is expected to do well" and "what exactly does the proposal do to solve it [problem of multivariate conditioning]":*
>
> As we mentioned in the introduction, the current learning approaches (that are implementable) for graphical event models are primarily score-based and they are all parametric, i.e. assume a very restricted intensity function model class. Constraint-based algorithms in Bayesian networks have been shown to have several advantages (including handling confounders) that are well known in the literature. Our intuition is that conducting statistical tests for process independence can provide similar benefits while doing multivariate conditioning in event streams; this has also been proposed at a high-level by others (e.g. Meek 2014) but we are unaware of prior empirical demonstrations in this regard, for event streams.
>
> Our line of inquiry here focuses on intensity functions being a real valued function on the Boolean domain; these are a member of a universal model class involving Boolean binning over the time line (Gunawardana and Meek, 2016). We therefore view this work as trying to realize process independence based learning of GEMs, i.e. the vision of prior work using testers designed for a member of a universal intensity function class.
>
> *Regarding "how should we expect influence tester to compare with likelihood ratio tester? ... when do we expect each of them to perform well?":*
>
> We expected the influence tester to perform better with these datasets as it leverages the proximal assumption in the graphical event models that were studied. The likelihood ratio tester is more general and therefore we expect it to be more robust over a broader class of models (although that remains to be demonstrated). However, the results do not suggest much difference between the testers. We will add more discussion about this in Section 4, as the reviewer suggests. While we do not have theoretical results for the proposed testers, we feel the ideas and the empirical investigation in this paper make for initial contributions in this space, and hope it will spur further research on other suitable testers. Even in i.i.d (non-event) settings, it is very hard to come across sample complexity results for non-parametric conditional independence testers (unless strong assumptions are made).

---

### Decision · Program_Chairs · 2022-01-12

**Decision:**

Accept (Poster)

**Comment:**

The paper addresses an important question in analyzing event data, and the reviewers agree that the paper makes a solid technical contribution. I would encourage the authors to take the reviewer comments to heart and make the presentation a bit more accessible to a community whose core familiarity is with standard DAGs and iid data (acknowledging space constraints). Otherwise, the paper connects well with prior research in this area, and is a nice contribution.